# Identification and Analysis of WRKY Transcription Factors in Response to Cowpea Fusarium Wilt in Cowpea

**DOI:** 10.3390/plants13162273

**Published:** 2024-08-15

**Authors:** Yali Hao, Rui Liu, Zhenchuan Mao, Qihong Yang, Shijie Zheng, Xiaofei Lu, Yuhong Yang, Bingyan Xie, Jianlong Zhao, Yan Li, Guohua Chen, Jian Ling

**Affiliations:** 1College of Horticulture, Shanxi Agricultural University, Jinzhong 030810, China; haoyali2021@163.com; 2State Key Laboratory of Vegetable Biobreeding, Institute of Vegetables and Flowers, Chinese Academy of Agricultural Sciences, Beijing 100081, China; 18404983798@163.com (R.L.); maozhenchuan@caas.cn (Z.M.); yqh20000416@163.com (Q.Y.); zheng200024@163.com (S.Z.); yangyuhongcaas@163.com (Y.Y.); xiebingyan@caas.cn (B.X.); zhaojianlongcaas@163.com (J.Z.); liyancaas1983@163.com (Y.L.); 3Institute of Zhongnong Tuba, Beijing 100081, China; luxiaofeicaas@163.com

**Keywords:** cowpea, cowpea Fusarium wilt, WRKY, *Fusarium oxyporum*, transcription regulation, expression profiling

## Abstract

In plants, WRKY transcription factors play a crucial role in plant growth, development, and response to abiotic and biotic stress. Cowpea (*Vigna unguiculata*) is an important legume crop. However, cowpea Fusarium wilt (CFW), caused by *Fusarium oxysporum f. sp. tracheiphilum* (Fot), poses a serious threat to its production. In this study, we systematically identified members of the cowpea WRKY (VuWRKY) gene family and analyzed their expression patterns under CFW stress. A total of 91 WRKY transcription factors were identified in the cowpea genome. Phylogenetic and synteny analyses indicated that the expansion of VuWRKY genes in cowpea is primarily due to recent duplication events. Transcriptome analysis of cowpea inoculated with *Fo* revealed 31 differentially expressed VuWRKY genes, underscoring their role in the response to CFW infection. Four differentially expressed WRKY genes were selected for validation. Subcellular localization and Western blot assays showed their nuclear localization and normal expression in *N. benthamiana*. Additionally, yeast one-hybrid assays demonstrated that VuWRKY2 can bind to the promoter region of the Catalase (CAT) gene, indicating its potential role in transcriptional regulation. This study establishes a foundation for further exploration of the role and regulatory mechanisms of VuWRKY genes in response to CFW stress.

## 1. Introduction

The WRKY transcription factors play a crucial regulatory role in various processes, including plant growth, development, dormancy, lignin synthesis regulation, and stress responses [1]. Members of the WRKY transcription factor family possess a highly conserved domain consisting of 60 amino acid residues, known as the WRKY domain. This domain includes a conserved N-terminal sequence, WRKYGQK, and a zinc finger structural motif (C-X4-5-C-X22-23-HXH) [2,3]. Based on the number of WRKY domains and the distinct characteristics of the zinc finger structure, WRKY proteins are classified into three groups. Group 1 contains two WRKY domains, while Groups 2 and 3 each have only one WRKY domain. Both Group 1 and Group 2 share the common C2H2 zinc finger structure, whereas Group 3 has a unique C2HC zinc finger structure [4,5]. The amino acid sequence diversity within Group 2 WRKY members further divides them into five subgroups: 2a, 2b, 2c, 2d, and 2e [6,7]. Since the identification of the first WRKY transcription factor (SPF1) in sweet potatoes in 1994 [8], the WRKY transcription factors family has been characterized in numerous species through whole-genome sequencing, enhancing our understanding of this gene family. For instance, *Arabidopsis thaliana*, *Oryza sativa*, *Glycine max*, and *Raphanus sativus* genomes contain 74, 109, 197, and 126 WRKY gene family members, respectively [9,10,11,12].

Extensive research indicates that members of the WRKY transcription factor family play a crucial role in responding to various abiotic and biotic stresses [13]. During their growth and development, plants face numerous biotic stresses, with pathogen attacks posing a significant threat. WRKY transcription factors are actively involved in the plant’s response to these biological stresses. Birkenbihl et al. [14] demonstrated that Arabidopsis *AtWRKY33* regulates the expression of many disease resistance-related genes. Furthermore, Arabidopsis *AtWRKY33* is induced upon infection by pathogenic organisms, thereby enhancing plant resistance to *Alternaria raphani Groves* and *Botrytis cinerea* [14,15]. Overexpression of the *CaWRKY40* gene in chili peppers affects hypersensitive response (HR)-related genes and pathogen-related genes, which enhances resistance to *Ralstonia solanacearum* [16]. Cheng et al. [17], through overexpression and RNAi techniques, demonstrated that *WRKY13*, *WRKY42*, and *WRKY45* participate in the resistance response of rice to *Magnaporthe oryzae*. *WRKY42* negatively regulates resistance to *Magnaporthe oryzae* by suppressing the expression of jasmonic acid (JA) signaling-related genes. WRKY transcription factors can also interact with other transcription factors to collectively inhibit pathogen infection. For example, the red-skinned pears employ the interaction between *PyWRKY26* and *PybHLH3* to target and activate the *PyMYB114* promoter. This leads to anthocyanin accumulation and enhanced resistance to *Aspergillus niger* [18].

Increasing evidence suggests that WRKY transcription factors play a crucial role in Fusarium wilt resistance by regulating the transcription of defense-related genes. In *Lilium regale*, *LrWRKY1* regulates the expression of the resistance gene *LrPR10-5*, participating in the defense response against *Fusarium oxysporum* [19]. *LrWRKY4* positively regulates the immune response in *Lilium regale*, potentially through involvement in JA/SA-mediated signaling pathways, inducing the expression of defense-related genes, and thereby regulating *Lilium regale*’s resistance to *Fusarium oxysporum* [20]. In chickpea, the transcription factor *CaWRKY40* binds to the promoter regions of two defense-related genes, CaDefensin and *CaWRKY33*, and positively regulates their transcription. Overexpression of *CaWRKY40* increases the transcription levels of *CaDefensin* and *CaWRKY33*, thereby enhancing resistance to *Fusarium oxysporum* [21]. The flax transcription factor *LuWRKY36* plays a key regulatory role in the defense response against *Fusarium oxysporum* by mediating hormone and calcium signaling pathways. Additionally, *LuWRKY36* positively regulates the expression of the *LuPLR1* gene and lignin biosynthesis under biotic stress [22]. Despite the completion of cowpea whole-genome sequencing, the transcriptome has not been thoroughly investigated. Currently, there are very few reports on cowpea transcription. Santos [23] identified candidate genes for resistance to root-knot nematodes through a combination of QTL mapping and transcriptome analysis. Tan [24] used RNA-seq technology to analyze the differential gene expression of two cowpea varieties with different cold tolerances under low-temperature stress. However, there is no reported research on WRKY transcription factors in cowpeas responding to Fusarium wilt.

Cowpea is a globally important vegetable due to its high nutritional value and economic returns. In China, cowpeas are cultivated across various regions, especially in tropical areas, with a cultivation area of approximately 5340 hectares. It is one of the primary vegetables grown in southern Hainan for winter cultivation and is transported to northern regions [25]. However, continuous monoculture has led to severe outbreaks of diseases and pests, with *Fusarium oxysporum* being the most critical, causing a 70% reduction in cowpea yield [26]. Cowpea Fusarium wilt is a significant disease caused by the vascular specialized form of *Fusarium oxysporum f. sp. tracheiphilum*. Currently, the pathogen is reported to be classified into four races [27]. After being infected by Fusarium wilt, cowpeas exhibit symptoms such as leaf discoloration, yellowing, leaf drop, wilting, vascular browning, and death. Research on cowpea Fusarium wilt has primarily focused on pathogen identification and control, with no studies currently reported on the mechanisms underlying cowpea’s response to Fusarium wilt. Moreover, there is no literature reporting on the WRKY gene response to cowpea Fusarium wilt infection.

While WRKY transcription factors have been identified in cowpea [28], their role in response to pathogen infection, particularly *Fusarium oxysporum* infection, has not been reported. In this study, we identified and analyzed the expression of potential WRKY genes using the cowpea genome as a reference, which spans 518.6 Mb and consists of 11 chromosomes. We identified 31 differentially expressed WRKY genes in response to cowpea Fusarium wilt. Among these, four genes, as confirmed by subcellular localization and Western blot experiments, were found to be localized in the cell nucleus and expressed normally in *N. benthamiana* leaves. The binding of VuWRKY2 to the promoter region of the CAT gene suggests its potential involvement in transcriptional regulation. These findings lay the foundation for further research into the mechanisms underlying cowpea’s response to Fusarium wilt. They provide data support and a theoretical basis for investigating the role of this transcription factor family in cowpea’s response to Fusarium wilt infection and stress conditions, as well as their role in biotic stress defense and functional validation.

## 2. Results

### 2.1. Genome-Wide Identification of WRKY Genes in Cowpea

In a previous study, 92 WRKY genes were identified in the cowpea genome (v1.1) [28]. Although both that study and the present one used the IT97K-499-35 genome, the previous study utilized data from the Phytozome database, whereas this study used data from NCBI. To identify WRKY genes in the cowpea genome, we conducted a Hidden Markov Model (HMM) search using the WRKY domain (PF03106). This approach identified a total of 91 WRKY genes across the cowpea genome. Subsequent analysis with SMART v9.0 and PFAM v36.0 software confirmed the presence of the WRKY domain in these genes. All VuWRKY genes are distributed across the 11 chromosomes of cowpea. Based on their chromosomal locations, the genes were sequentially named VuWRKY1 to VuWRKY91. The results of chromosomal localization analysis (Figure 1) revealed that the 91 VuWRKY genes are unevenly distributed across all 11 chromosomes. Chromosome 3 (Chr3) exhibited the highest number of distributed VuWRKY genes, with a total of 18, while Chromosomes 4 (Chr4) and 12 (Chr12) had the fewest, each hosting only 1 VuWRKY gene.

Comparison with the previously identified VuWRKY genes revealed discrepancies: one fewer gene on chromosomes 7, 8, and 9, respectively, and two additional genes on chromosome 10. The nomenclature also differs, with corresponding genes being renamed in this study (Appendix A). These discrepancies are likely due to differences in the genome versions used and the methods employed for WRKY gene identification.

Additionally, WRKY transcription factor family genes were identified in Glycine max (Appendix A), and the evolutionary relationships among *Arabidopsis thaliana* WRKY (AtWRKYs), *Glycine max* WRKY (GmWRKYs), and VuWRKYs were analyzed. Phylogenetic analysis revealed that VuWRKYs share a close evolutionary relationship with AtWRKYs and GmWRKYs (Appendix A).

A comprehensive analysis of the physicochemical properties of cowpea VuWRKY members was also conducted (Appendix A). Among them, VuWRKY17 has the longest amino acid sequence (1376 aa), while VuWRKY73 has the shortest (131 aa). The isoelectric points (pl) of these proteins range from 4.79 to 9.94. The molecular weights of these proteins vary from 18,201.76 Da to 80,985.69 Da. Predictions of protein stability indicate that only six VuWRKY proteins have instability coefficients below 40, suggesting they are stable, while the remaining proteins are considered unstable. Subcellular localization predictions reveal that 85 VuWRKY proteins are located in the nucleus, 4 VuWRKY proteins are in the chloroplasts, VuWRKY81 is positioned in the cytoplasm, and VuWRKY17 is localized in the plasma membrane.

### 2.2. Phylogenetic Tree Construction of VuWRKY Family Members and Analysis of Conserved Protein Domains in VuWRKY Proteins

To assess the evolutionary relationships among VuWRKY genes, a phylogenetic tree was constructed using VuWRKY and *Arabidopsis thaliana* WRKY (AtWRKY) genes. As shown in Figure 2, the VuWRKY gene family can be classified into three groups (Groups 1–3). Group 1 consists of 15 members, each containing two WRKY domains, and is further subdivided into Group 1N and Group 1C. Group 2 consists of 60 VuWRKY members, which are further divided into five subgroups (Group 2a to Group 2e) with 5, 15, 20, 8, and 12 members, respectively. The remaining 16 VuWRKY members are categorized into Group 3.

The results of the conserved protein domain analysis indicate that the VuWRKY domain is relatively conserved, with some specific variations (Appendix A). Notably, in Group 1, the conserved WRKYGQK motif at the N-terminus of VuWRKY84 changes to WRKYGEK. Similarly, in Group 2c, the conserved WRKYGQK motif of VuWRKY14 transforms into WRKYGEK. The WRKYGQK motif of AtWRKY50 shifts to WRKYGKK, and similarly, VuWRKY16, VuWRKY58, VuWRKY73, VuWRKY75, and VuWRKY81 exhibit a change from WRKYGQK to WRKYGKK, suggesting that these variations might confer new biological functions. Regarding zinc finger motifs, most VuWRKY proteins exhibit the C2H2 type, except those in Group 3, which have the C2HC type. Among the 91 VuWRKY genes in cowpea, the C2H2 type zinc finger motifs in Group 1N are CX4C22HXH, while Group 1C’s zinc finger structure is CX4CX23HXH. The zinc finger motifs for Group 2a and Group 2b are CX5C23HXH, and for Group 2c, 2d, and 2e, the C2H2 type zinc finger motifs are CX4C23HXH, CX6C23HXH, and CX5C23HXH, respectively. Group 3 is characterized by the C2HC-type zinc finger motif CX7C23HXC. Additionally, the zinc finger motif for Group 2d is C2H2, but VuWRKY39 undergoes a mutation, changing from the original C2H2 to C2HC. In Group 3, the VuWRKY59 zinc finger structure undergoes a mutation from C2HC to C2H2 (Appendix A).

### 2.3. Analysis of the Gene Structure and Conserved Motifs of Cowpea WRKY Genes

The diversity in gene structure serves as a crucial criterion for classifying gene families (Figure 3). The positional information of exons and introns was obtained from the annotation file of the cowpea genome. The number of introns varies significantly among different VuWRKY genes, ranging from 1 to 6. VuWRKY28 and VuWRKY56 have the highest number of introns, with both possessing 6. Six VuWRKY genes possess only one intron. The majority, 48 genes, contain two introns. Notably, VuWRKY87 and VuWRKY43 genes lack untranslated region (UTR) structures.

At the MEME online site, we configured the analysis for 10 conserved motifs across the 91 cowpea WRKY proteins. As shown in Figure 4, the VuWRKY gene family contains between 2 and 8 conserved motifs, with all 91 genes containing Motif1 and Motif2. This suggests that Motif1 and Motif2 are core conserved motifs within the VuWRKY family. In Group 1, 15 VuWRKY proteins harbor two WRKYGQK motifs, similar to Motif1. Motif3 also contains the WRKYGQK motif, indicating functional similarity between these two conserved motifs. However, in Group 2b, the VuWRKY9 protein also contains Motif3, hinting at a possible mutation during genetic evolution. Except for VuWRKY43 and VuWRKY86, all other proteins contain conserved motifs. Additionally, certain motifs are specific to particular groups, such as Motif5, which is predominantly present in Group 1, Group 2b, and Group 2c; Motif6 and Motif9, mainly found in Group 2b; and Motif10, primarily existing in Group 2d and Group 2e. These findings suggest that similar conserved motifs are present in VuWRKY members of the same group while different types of VuWRKY family members contain distinct conserved motifs, indicating that a specific motif may play a role in different types of VuWRKY family members.

### 2.4. Homology Analysis of VuWRKY Genes

To further explore the amplification and evolution of VuWRKY genes, we conducted a genome-wide analysis of gene duplications within the VuWRKY gene family. We identified 46 recent duplication events involving 55 VuWRKY genes (Figure 4A). However, no tandem duplication events were observed among the VuWRKY genes. These findings suggest that some VuWRKY genes may have arisen from segmental duplication events, indicating that segmental duplications play a crucial role in the evolution of the VuWRKY gene family.

To investigate the potential evolutionary relationships among VuWRKY genes, we performed comparative syntenic analysis using VuWRKY, GmWRKY, and AtWRKY genes. A total of 125 soybean genes and 66 Arabidopsis genes exhibited syntenic relationships with VuWRKY genes (Figure 4B). Among these, 59 conserved syntenic relationships were identified among the cowpea, soybean, and Arabidopsis genomes, indicating the conservation of most WRKY genes across these three species.

### 2.5. Expression Patterns of VuWRKY Genes in Response to Cowpea Fusarium Wilt Infection

To gain preliminary insights into the role of the cowpea WRKY gene family in Fusarium wilt resistance, transcriptome data from cowpea plants inoculated with Fusarium wilt pathogen for 12 days were analyzed for both the treatment and control groups to assess the expression of 91 WRKY genes. Of these genes, 31 exhibited differential expressions. Based on their transcript per million (TPM) values, analysis of the expression patterns of these 31 VuWRKY members in response to Fusarium wilt pathogen infection in resistant cowpea varieties revealed the following (Figure 5): 27 VuWRKY genes were upregulated, while 4 VuWRKY genes were downregulated (*p* < 0.01). Specifically, within WRKY Group 1, 3 VuWRKY genes showed differential expressions; in Groups 2a to 2b, there were 1, 5, 10, 1, and 6 VuWRKY genes showing differential expression, respectively; and in Group 3, 5 genes showed differential expression. The differential expression of these 31 VuWRKY genes indicates their involvement in responding to Fusarium wilt infection.

RT-qPCR analysis of VuWRKY gene expression under Fusarium wilt stress revealed distinct expression patterns. At 0, 5, 12, 17, and 22 days post-inoculation, cowpea plants displayed varying symptoms, and disease severity was recorded (Figure 6A and Appendix A). Based on the distribution of cowpea VuWRKY genes across different WRKY groups, 11 significantly differentially expressed genes were selected (Figure 6B). Among these, 8 VuWRKY genes reached their highest expression levels at 12 days. VuWRKY37, VuWRKY41, VuWRKY59, and VuWRKY77 exhibited an initial upregulation followed by downregulation, whereas VuWRKY2, VuWRKY22, VuWRKY49, VuWRKY55, and VuWRKY70 reached their highest expression levels at 12 days, followed by an initial downregulation and subsequent upregulation. Additionally, VuWRKY20 and VuWRKY52 exhibited a downward trend in expression at 12 days. Notably, VuWRKY2, VuWRKY22, VuWRKY37, VuWRKY41, VuWRKY49, and VuWRKY70 demonstrated rapid expression changes compared to the control group, indicating swift activation of immune defense responses. Furthermore, VuWRKY59 exhibited a sharp increase in expression levels at 5 days in the control group compared to day 0, which was significantly higher than in the treatment group; however, by day 12, expression levels in the control group declined rapidly, while the treatment group showed a peak, which warrants further investigation.

### 2.6. Subcellular Localization of VuWRKY Proteins

In this study, we investigated the responses of the VuWRKY2, VuWRKY22, VuWRKY49, and VuWRKY59 genes to infection by the cowpea wilt disease. WoLF PSORT predicted that the proteins encoded by these genes are localized in the cell nucleus. To validate the prediction, GFP-fusion protein vectors, namely super1300-VuWRKY2-GFP, super1300-VuWRKY22-GFP, super1300-VuWRKY49-GFP, and super1300-VuWRKY59-GFP, were transiently expressed in *N. benthamiana* leaves. DAPI staining was used as a nuclear localization control, while super1300-GFP served as an empty vector control. Confocal microscopy was employed to observe the co-localization of signals from green fluorescent protein (GFP) with DAPI nuclear signals, confirming that the four VuWRKY proteins are localized in the nucleus, consistent with the predicted localization (Figure 7).

Western blot analysis of cowpea WRKY proteins in *N. benthamiana* leaves (Figure 8A) demonstrated successful expression of the VuWRKY proteins. The proteins exhibited the expected size when fused with the super1300-GFP vector, indicating that the Super-VuWRKYs-GFP overexpression vector was efficiently introduced into the target plants and expressed as intended.

### 2.7. Binding of VuWRKY2 to the CAT Promoter

To validate the potential interaction between VuWRKY2 and the CAT (DEO72_LG7g45) promoter, VuWRKY2 was fused with the pB42AD vector, and the CAT promoter was linked to the pLaczi vector for co-transformation in EGY48. Subsequently, streaking was performed on SD/-Ura/-Trp plates and X-gal-containing SD/-Ura/-Trp plates. Notably, a blue coloration was observed on the SD/-Ura/-Trp plates containing X-gal, which is consistent with the positive control and indicated a successful interaction between VuWRKY2 and the CAT promoter. In contrast, the negative control did not show a blue coloration, further confirming that the interaction between VuWRKY2 and the CAT promoter is specific and not due to non-specific binding. These results provide evidence for the binding between VuWRKY2 and the CAT promoter, indicating a potential regulatory relationship between VuWRKY2 and the target gene (Figure 8B).

## 3. Discussion

WRKY transcription factors are unique to plants, participating in the regulation of plant growth and development. Simultaneously, in response to external environmental stress, plants exhibit corresponding responses [29,30,31,32]. The WRKY gene family has been identified in several species, such as *Arabidopsis thaliana* (74 genes) [33], rice (*Oryza sativa*) (102 genes) [10], maize (*Zea mays*) (52 genes) [34], soybean (*Glycine max*) (182 genes) [35], and alfalfa (*Medicago sativa*) (198 genes) [36]. The number of WRKY genes in soybean and alfalfa is significantly higher than in maize, which is attributed to both segmental and tandem duplications in these species. For example, alfalfa exhibits 351 gene duplication events, consisting of 332 segmental duplications and 19 tandem duplications. However, in this study, WRKY gene duplication events in cowpeas are consistent with maize. Among the 55 identified VuWRKY genes, 46 are the result of segmental duplications, with no tandem duplications observed. This suggests that segmental duplications are the primary drivers of WRKY gene family expansion in cowpea. A systematic study on the cowpea WRKY gene family has not been reported. Based on cowpea genome data, this study identified 91 WRKY members. By constructing a phylogenetic tree with Arabidopsis, the VuWRKY family was classified into three subfamilies, with Group 2 further divided into 2a, 2b, 2c, 2d, and 2e. This classification is consistent with other research results [6].

Conservative domain analysis reveals that most VuWRKY domains are relatively conserved, though some protein domains have undergone variations. For example, VuWRKY84N and VuWRKY14 have mutated from WRKYGQK to WRKYGEK, while VuWRKY16, VuWRKY58, VuWRKY73, VuWRKY75, and VuWRKY81 have mutated from WRKYGQK to WRKYGKK. Similar variations are observed in other plants, such as the WRKYGKK motif in strawberry [37] and carnation [38]. Variations similar to WRKYGKK are found in rice [39], barley [40], and quinoa [41]. In the WRKY proteins of *Kandelia candel* [42], the core domains of KcWRKY14 and KcWRKY27 members of the WRKY family have undergone variations to WRKYGEK and WRKYGKK, respectively, showing similar variations in conservative domains. Studies indicate that conserved domains are crucial for the precise binding of WRKY transcription factors to the W-box on target genes. Variations in the WRKYGQK heptapeptide core sequence can significantly affect plant functions. For instance, the GKK mutant type in soybeans loses its ability to bind to the W-box, while the GKK mutant type in tobacco can specifically recognize the WK box. When GKK is mutated to GQ(E)K, this specific binding to the WK box is lost [43,44]. Therefore, it can be inferred that VuWRKY gene family members with mutations in the heptapeptide domain may specifically recognize other cis-elements outside the W-box, resulting in novel functionalities.

Among the conserved domain variations mentioned above, except for VuWRKY84N, the remaining variations are distributed within Group 2c. Increasing evidence suggests that Group 2c WRKY transcription factors are involved in plant immune responses. In Arabidopsis, *WRKY75* has been shown to positively regulate resistance to *Botrytis cinerea* by modulating the JA-mediated signaling pathway [45]. *WRKY48* is identified as a negative regulator of basal defense against the bacterial pathogen *Pseudomonas syringae* by downregulating pathogen-induced expression of pathogenesis-related genes [46]. Gao et al. also demonstrated that *WRKY50* and *WRKY51* positively regulate resistance to downy mildew in Arabidopsis through the SA-mediated signaling pathway [47]. These studies collectively suggest a crucial role of Group 2c WRKY transcription factors in plant immunity. In cotton, the Group 2c transcription factor enhances resistance against *Fusarium oxysporum* by inducing the GhMKK2-GhNTF6-GhMYC2 pathway, which increases the accumulation of flavonoids [48]. However, there are no reports of WRKY transcription factors specifically responding to cowpea Fusarium wilt. This study observed that the expression levels of 31 VuWRKY genes changed following infection by the cowpea Fusarium wilt pathogen, indicating their involvement in response to Fusarium wilt stress in cowpea. The expression intensity of these 31 differentially expressed VuWRKY genes varied, and many of these genes have homologous genes involved in plant immune responses. For example, the homologous gene of *VuWRKY2*, *AtWRKY75*, may form complexes with ET signaling or other defense-related proteins to regulate defense responses against necrotrophic pathogens [45]. *AtWRKY75* also positively regulates resistance to *Sclerotinia sclerotiorum* [49]. The homologous genes *VuWRKY22*, *AtWRKY4*, and *AtWRKY3* play positive regulatory roles in the resistance response to saprophytic pathogens, with single mutants and double mutants showing reduced resistance to *Botrytis cinerea*. However, *AtWRKY4* overexpression plants also exhibit increased susceptibility to *Pseudomonas syringae* [50]. The homologous gene of *VuWRKY49*, *AtWRKY25*, negatively regulates resistance to *Pseudomonas syringae* [51]. The double mutants of *VuWRKY59*’s homologous genes *AtWRKY18* and *AtWRKY60* show increased resistance to leaf spot pathogens and decreased resistance to *Botrytis cinerea* [52]. In this study, the cowpea Fusarium wilt pathogen, along with *Sclerotinia sclerotiorum* and *Botrytis cinerea*, are all fungi, indicating that the VuWRKY genes homologous to AtWRKY genes may have similar functions in response to fungal infections.

In this study, subcellular localization and Western blot analysis of VuWRKY2, VuWRKY22, VuWRKY49, and VuWRKY59 revealed that these four VuWRKY genes are located in the nucleus and are normally expressed in *N. benthamiana* leaves. Significant differential expression of these genes was observed at the early stages of infection, suggesting that the differential expression of VuWRKY genes is associated with the response to cowpea Fusarium wilt infection. This is the first report of WRKY genes responding to cowpea Fusarium wilt infection in cowpea. Moreover, VuWRKY2 was found to bind to the CAT promoter region, potentially participating in transcriptional regulation. These findings provide valuable insights for further studies on the role of VuWRKY genes in the resistance mechanism of cowpea to Fusarium wilt. Consequently, this study advances our understanding of how WRKY genes regulate plant responses to Fusarium wilt infection. However, further research is needed to elucidate the underlying mechanisms, laying the groundwork for breeding cowpea with enhanced disease resistance.

## 4. Materials and Methods

### 4.1. Genome-Wide Identification of VuWRKY Genes in Cowpea

The cowpea whole genome data were downloaded from the NCBI database (http://www.ncbi.nlm.nih.gov/, accessed on 28 May 2023), and the HMM file for the WRKY conserved domain protein sequences (sequence number PF03106) was obtained from the Pfam database (http://pfam.xfam.org/, accessed on 10 June 2023). WRKY gene identification was conducted on a Linux system using HMMER (http://hmmer.org/, accessed on 10 June 2023), setting the E-value threshold to 1 × 10^−20^ to obtain candidate genes (parameters: hmmsearch-E 1 × 10^−20^ WRKY.hmm cowpeaprotein.fa > out1.fa). Further validation of candidate genes for the presence of typical WRKY domains was conducted using SMART v9.0 software (http://smart.embl.de/, accessed on 10 June 2023) and Pfam v36.0 (http://pfam.xfam.org/search, accessed on 10 June 2023).

Physicochemical properties of WRKY proteins were analyzed using ExPASy (https://web.expasy.org/protparam/ accessed on 20 June 2023). Additionally, WoLF PSORT (https://www.genscript.com/wolf-psort.html?src=leftbar, accessed on 20 June 2023) was used to predict the cellular localization of WRKY proteins.

### 4.2. Gene Structure, Conserved Motifs, and Phylogenetic Tree Analysis

Multiple sequence alignments were generated using ClustalW v2.1 with default settings (https://www.genome.jp/tools-bin/clustalw, accessed on 18 November 2023), incorporating VuWRKY proteins and seven selected AtWRKY proteins.

Genomic and coding sequence (CDS) data for VuWRKY1-VuWRKY91 genes were retrieved from the NCBI genome annotation files. Conserved amino acid motifs in the cowpea VuWRKY1-VuWRKY91 sequences were analyzed using the MEME online tool (https://meme-suite.org, accessed on 18 November 2023), with a motif search set to identify 10 motifs, and the final positions for each gene were obtained. Structural characteristics of VuWRKY1-VuWRKY91 genes were illustrated using the GSDS website (http://gsds.gao-lab.org/, accessed on 27 November 2023). Protein sequences related to *Arabidopsis thaliana* were downloaded from the TAIR website. The iqtree v2.2.5 software was used to construct an evolutionary tree with 100 bootstrap repetitions, employing the maximum likelihood method. The classification of the cowpea WRKY transcription factor family was determined through evolutionary analysis. The tree’s visual representation was enhanced using iToL.

### 4.3. Chromosomal Localization, Gene Duplication, and Synteny Analysis of VuWRKY

The chromosome lengths of cowpea and the positional information of the VuWRKY gene family members were extracted from the cowpea genome GFF3 annotation file. VuWRKY genes were visualized on the chromosomes using MG2C v2.1 [53] (http://mg2c.iask.in/mg2c_v2.1/, accessed on 4 December 2023) software and renamed as VuWRKY1-VuWRKY91 based on their positions. Duplication events of the VuWRKY genes were identified using MCScanX with default parameters. Homology maps of VuWRKY with two species, soybean and *Arabidopsis thaliana*, were constructed using the Multiple Synteny Plot function of TBtools v2.119 software [54].

### 4.4. Plant Material and Cowpea Fusarium Wilt Pathogen Inoculation

The cowpea cultivar *Fengjiang1* is widely cultivated in southern China. The FOT strains were kindly provided by Rongfeng Xu from the culture collection center of Fujian Agriculture and Forestry University.

The inoculation method for cowpea Fusarium wilt was based on the protocol described by Lv et al. [55]. Inoculations were performed on cowpea seedlings at the two-leaf stage, ensuring uniform growth. The root systems were thoroughly rinsed with water and immersed in the pathogen solution for 15 min, while the control group underwent root immersion in sterile water. Subsequently, the seedlings were transplanted into sterilized plastic pots. Samples were collected at 0, 5, 12, 17, and 22 days post-inoculation to assess the disease index of cowpea plants following exposure to the Fusarium wilt pathogen. Three randomly selected root samples were taken for each treatment, with three replicates per group. The cleaned roots were rapidly frozen in liquid nitrogen and stored at −80 °C for subsequent use. Some samples were sent to Biomarker Technologies for transcriptome sequencing, while others were used for real-time quantitative PCR (qPCR).

### 4.5. RT-qPCR Transcriptome Analysis, Identification of Differentially Expressed Genes, and Expression Validation

Transcriptome data utilized cowpea roots inoculated with the Fusarium wilt pathogen as the treatment group, with water-treated cowpea roots serving as the control. The analysis of the expression profile of VuWRKYs during Fusarium wilt infection in cowpea was conducted 12 days post-inoculation. Cowpea roots were used for RNA-seq with three independent biological replicates. RNA-seq data processing and analysis of Differentially Expressed Genes (DEGs) were performed using the Salmon pipeline [56] and DESeq2 [57]. Genes were identified as DEGs based on an adjusted *p*-value < 0.01 and a log2FoldChange > 1, comparing the treatment group with the control at the same time point. 

Total RNA from the samples was extracted using the Polysaccharide Polyphenol Plant RNA Extraction Kit (TIANGEN). After electrophoresis detection, the extracted RNA was reverse transcribed into cDNA using the EvoM MLV RT Mix Tracking Kit with gDNA Clean for PCR (Yellow) and stored at −20 °C. The relative expression levels of VuWRKYs genes at different time points were detected using the CFX96 Real-Time PCR System (Bio-Rad). The cowpea *VuUbq28* (AT4G05320) [58] gene was selected as the reference gene, with the primer sequences: forward primer: GAGCTCAAGGACCTCCAGAA and reverse primer: CTAGAAAAACACCCCCAGCA. Data analysis was performed using the 2^−ΔΔCT^ method [59]. GraphPad Prism v8.0.2 software was used for variance analysis of the relative expression levels, and the GenesCloud tool was employed to generate a gene relative expression heatmap.

### 4.6. Subcellular Localization of VuWRKY and Western Blotting

The CDS of VuWRKY2, VuWRKY22, VuWRKY49, and VuWRKY59 were cloned into the super1300-GFP vector to generate super1300-VuWRKY2-GFP, super1300-VuWRKY22-GFP, super1300-VuWRKY49-GFP, and super1300-VuWRKY59-GFP constructs. These vectors were then transformed into *Agrobacterium tumefaciens* strain GV3101. *N. benthamiana* plants were grown for 4 weeks, and the transformed Agrobacterium GV3101 was adjusted to an OD600 of 0.5 in infiltration buffer (10 mM MgCl_2_, 10 mM MES, and 0.1 mM acetosyringone). After 48 h of injection, leaf imaging was conducted using an inverted confocal microscope (Zeiss LSM700, Jena, Germany) to capture GFP images at 488 nm.

The purified protein (100 µL, supplemented with 20 µL of 5× SDS-PAGE Sample Buffer) was boiled at 100 °C for 10 min, cooled on ice for 5 min, and then subjected to electrophoresis on an SDS-PAGE gel. The subsequent steps included transferring to a PVDF membrane, blocking (1× TBST containing 5% skim milk powder) for 1 h, incubating with the primary antibody for 1.5 h, washing with TBST (5 min per wash, 6 times), incubating with the secondary antibody for 1.5 h, and washing with TBST (5 min per wash, 6 times). Detection was performed using the EasySee Western Blot Kit. The antibodies used for labeling were anti-GFP and anti-H3 antibodies.

### 4.7. Yeast One-Hybrid

The Y1H experiment was conducted as described in previous studies [60]. Sequenced pB42AD and pLaczi vectors were transformed into the EGY48 yeast strain, which is capable of growing on SD/-Trp/-Ura medium. Validation of positive transformants involved selecting those that grew on media containing sucrose and 5-bromo-4-chloro-3-indolyl-β-D-galactopyranoside (X-gal), which exhibited visible color reactions.

## 5. Conclusions

The absence of resistance genes against cowpea Fusarium wilt in the cowpea genome has severely hindered cowpea cultivation. WRKY transcription factors are involved in the plant’s response to biotic stress. In this study, 91 WRKY genes were identified in the cowpea genome. Homology analysis indicated a high conservation of WRKY genes in cowpea compared to soybean and Arabidopsis. The expression patterns of VuWRKY genes were examined under cowpea Fusarium wilt stress. Subcellular localization and Western blotting confirmed that four VuWRKY genes are localized in the cell nucleus and are normally expressed in *N. benthamiana* leaves. This study represents the first report on the expression profile of WRKY genes in cowpea under Fusarium wilt stress. Additionally, VuWRKY2 was found to bind to the CAT promoter region, suggesting its potential involvement in transcriptional regulation. These findings provide important insights for further research into the resistance mechanisms of cowpea against cowpea Fusarium wilt.

## Figures and Tables

**Figure 1 plants-13-02273-f001:**
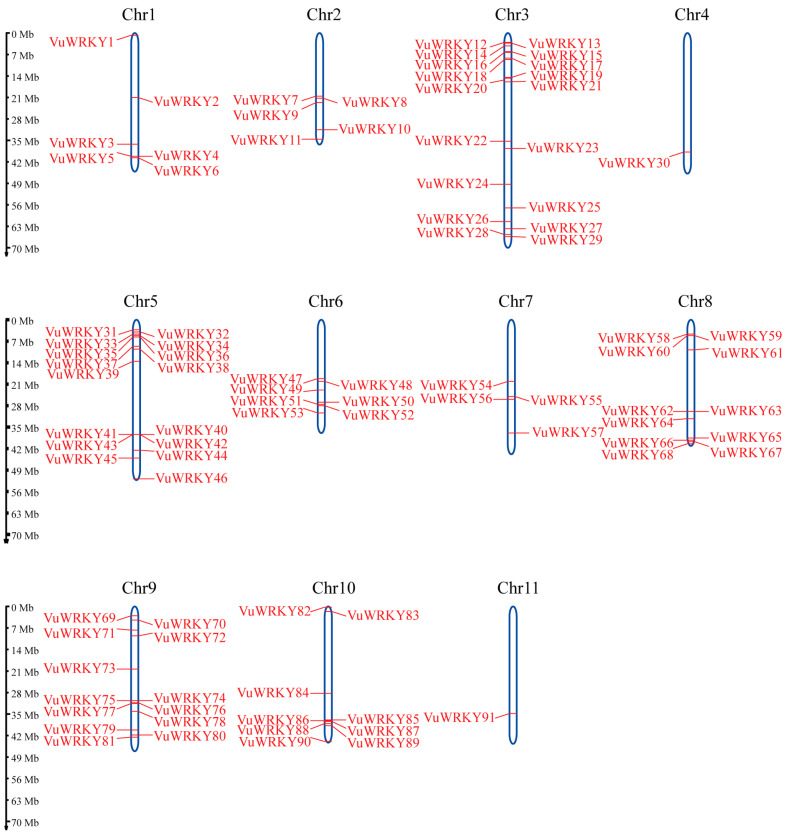
Visualization of the chromosomal location of the VuWRKY genes. Distribution of 91 VuWRKY genes on 11 chromosomes. Blue vertical lines depict chromosomes, with chromosome numbers located at the top of each. The left scale indicates chromosome length (Mb).

**Figure 2 plants-13-02273-f002:**
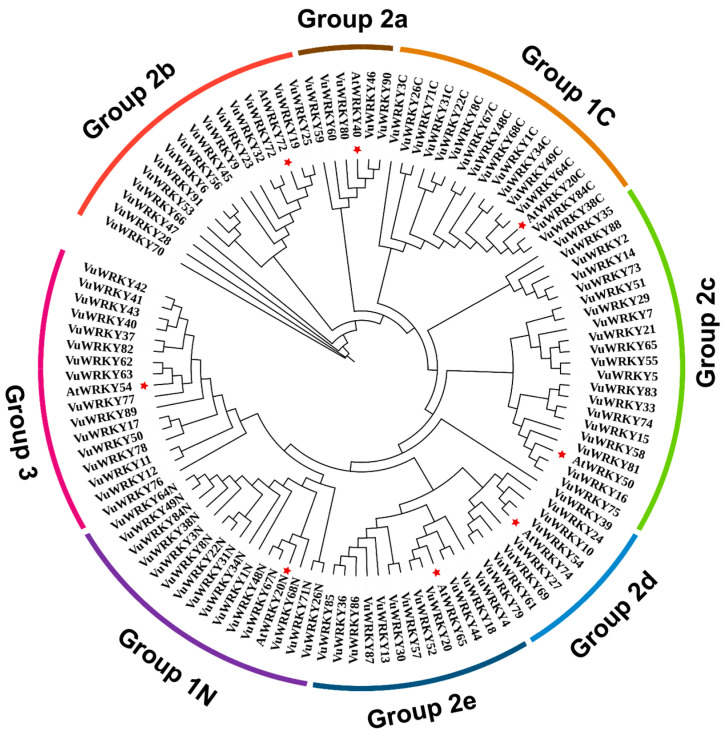
The phylogenetic tree of WRKY proteins from cowpea and Arabidopsis. The unrooted phylogenetic tree, depicting the relationships among WRKY family members in cowpea and Arabidopsis. Different colored arcs represent the eight subgroups of WRKY proteins. The red pentagrams denote WRKY genes in Arabidopsis, while the remaining symbols represent WRKY genes from cowpea.

**Figure 3 plants-13-02273-f003:**
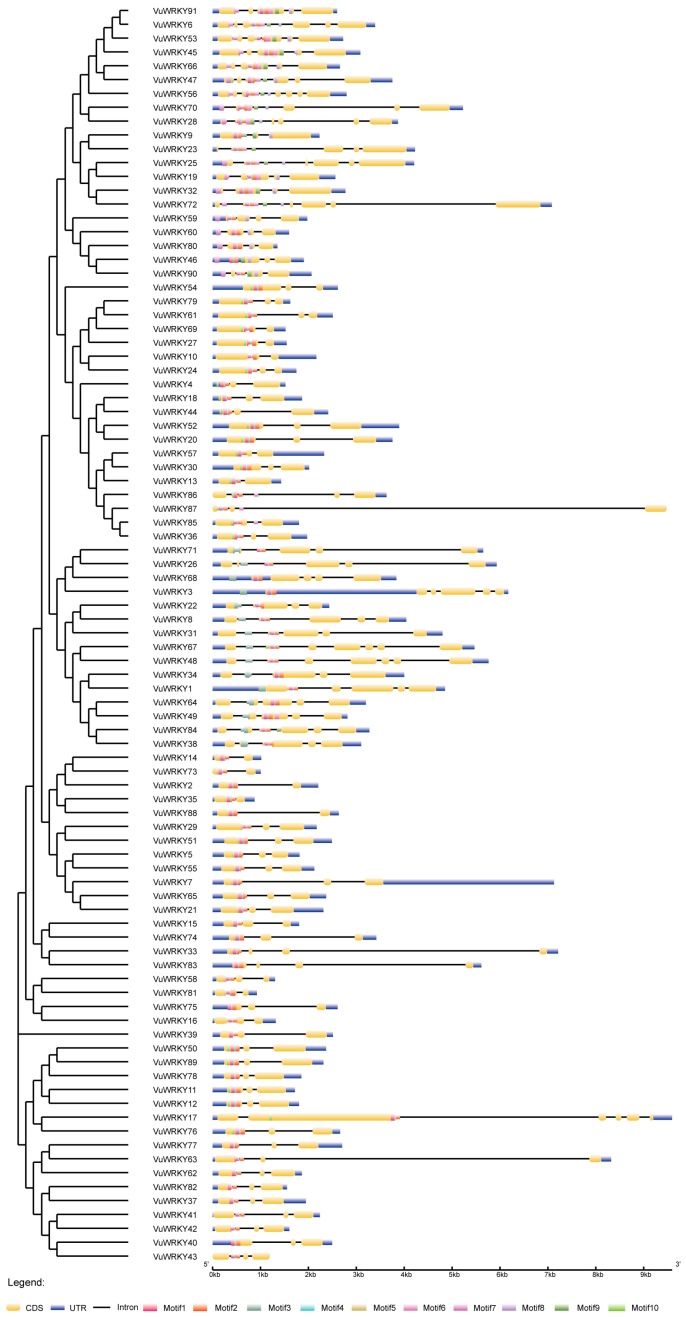
The phylogenetic tree relationship, conserved motifs, and gene structure of VuWRKY genes. The left panel illustrates the phylogenetic tree constructed from VuWRKY protein sequences. The right panel depicts conserved protein motifs and gene structure. Conserved motifs are represented by colored boxes labeled as motif1-motif10. In the gene structure, coding sequences (CDS) are indicated by yellow boxes, untranslated regions (UTR) by blue boxes, and introns by black lines.

**Figure 4 plants-13-02273-f004:**
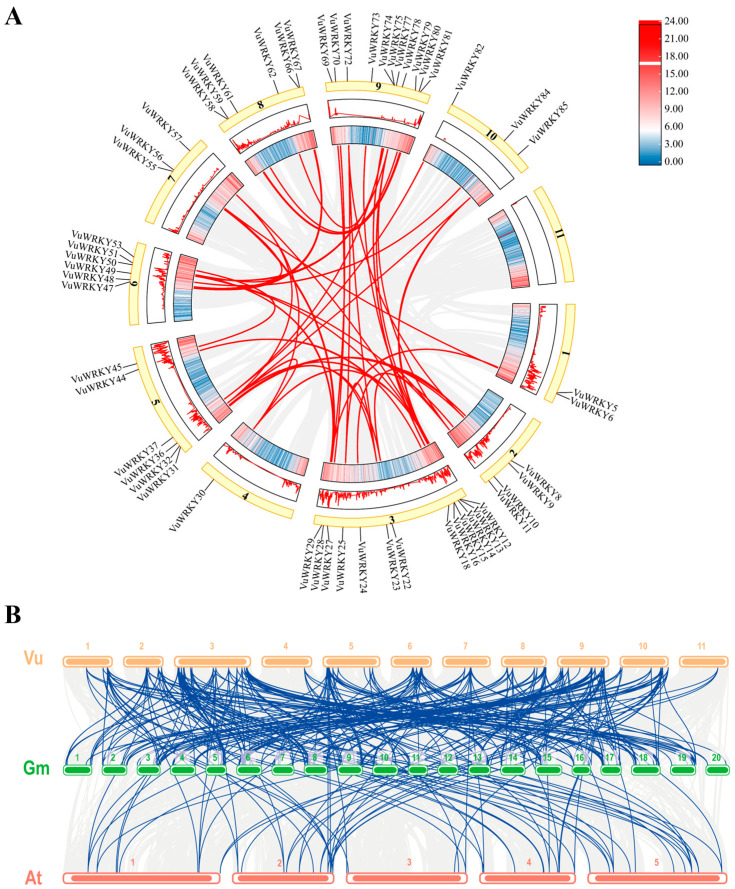
Collinearity analysis of the VuWRKY genes in cowpea. (**A**) Schematic representation of the chromosomal relationships among VuWRKY genes. Gray lines denote syntenic blocks within the cowpea genome, with duplicated WRKY gene pairs connected by red lines. (**B**) Synteny analysis of WRKY genes in cowpea, soybean, and Arabidopsis. Homologous blocks are indicated by gray lines, while syntenic gene pairs containing WRKY genes are highlighted with red lines. “Vu” represents cowpea (*Vigna unguiculata*), “Gm” represents soybean (*Glycine max*), and “At” represents *Arabidopsis thaliana*.

**Figure 5 plants-13-02273-f005:**
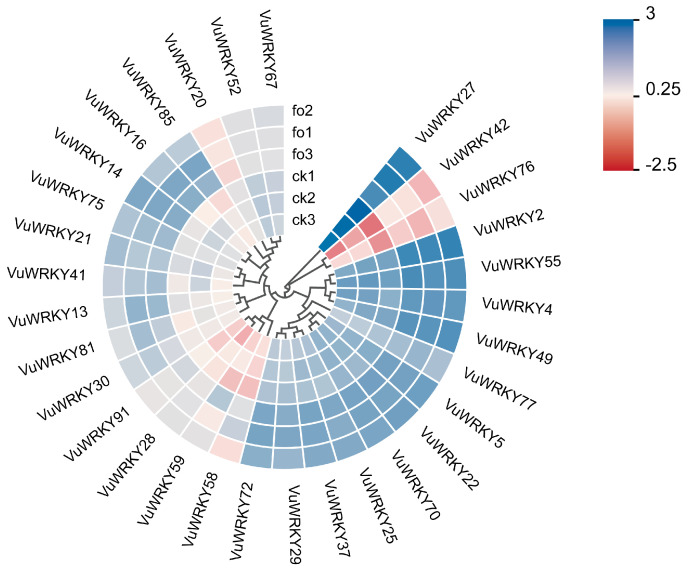
The expression pattern of WRKY genes in cowpea roots following infection with *Fusarium oxysporum* was investigated. Transcriptomic data were used to analyze the transcriptional levels of VuWRKY genes at 12 days post-infection. The heatmap displays the expression levels of VuWRKY genes using transcript per million (TPM) values. ck1, ck2, and ck3 represent control groups, while fo1, fo2, and fo3 represent the infected groups. The color scale ranges from red to blue, indicating increasing levels of expression.

**Figure 6 plants-13-02273-f006:**
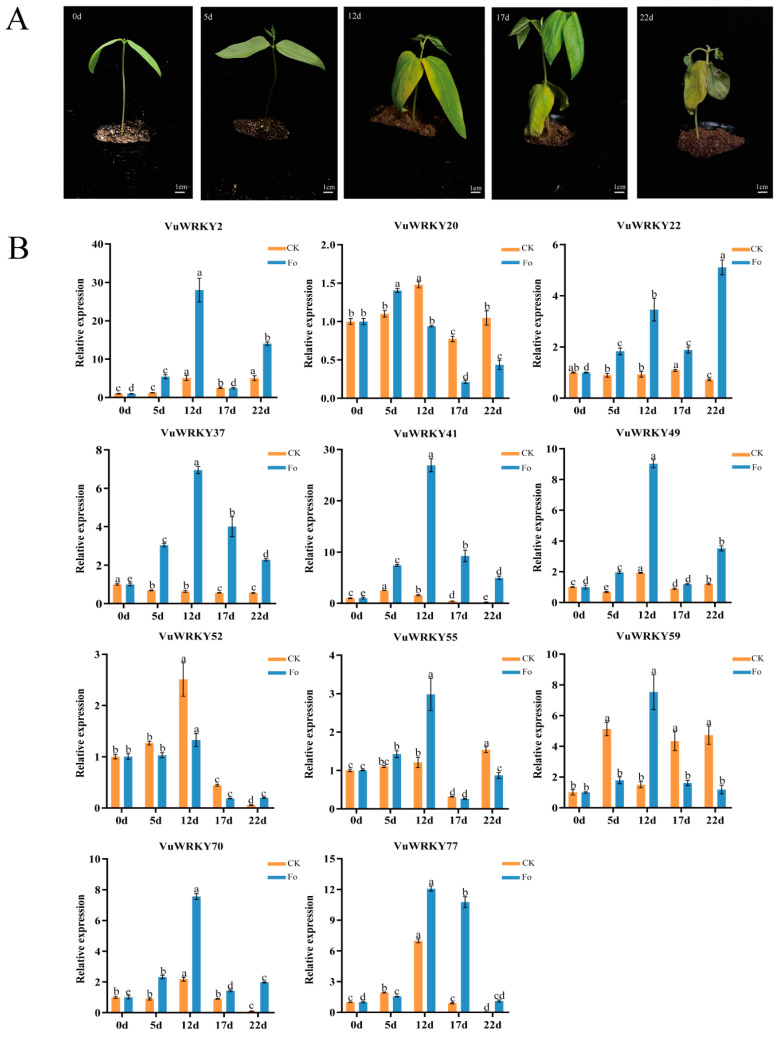
The phenotype and gene expression pattern analysis of cowpea Fusarium wilt infection. (**A**) The incidence of cowpea wilt pathogen on different days after infection of cowpea. Letters a, b, and c represent 0, 5, 12, 17, and 22 days of disease onset in cowpea plants, respectively. (**B**) The expression patterns of 11 VuWRKY genes in response to cowpea Fusarium wilt stress were analyzed. The relative expression levels (y-axis) were calculated based on the described method. Time points 0 days, 5 days, 12 days, 17 days, and 22 days represent the post-inoculation time (x-axis). Error bars were calculated based on three replicates. Letters represent significant differences between different days post-inoculation (*p* < 0.05).

**Figure 7 plants-13-02273-f007:**
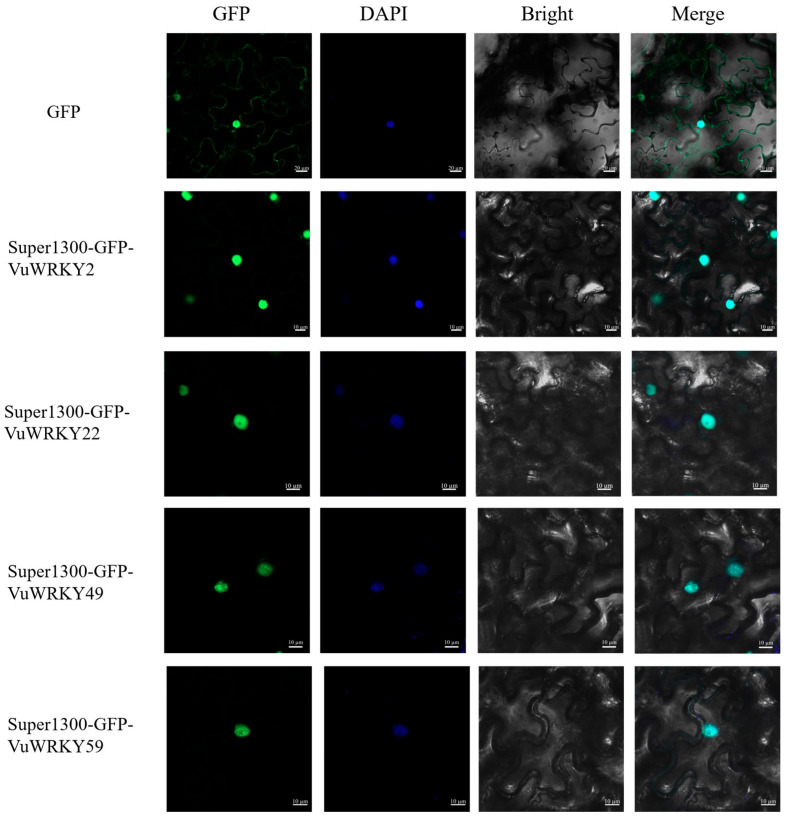
The laser channels included GFP fluorescence excitation, DAPI staining fluorescence, bright-field image, and merged image. *N. benthamiana* leaves were stained with DAPI 48 h post-infiltration. Gene localization within the leaf tissues was observed using a confocal laser scanning microscope.

**Figure 8 plants-13-02273-f008:**
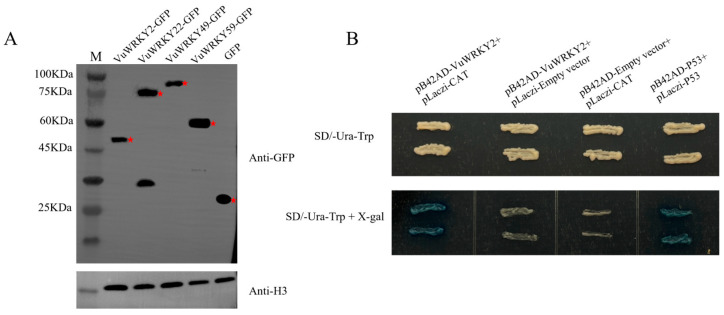
VuWRKY protein detection and VuWRKY2 binding to CAT promoter. (**A**) Western blot analysis was used to detect target proteins in VuWRKY-GFP and GFP. After transient expression in *N. benthamiana* leaf cells, the presence of VuWRKY-GFP in the nucleus was assessed using anti-GFP antibodies. Histone H3 was utilized as an internal control to confirm nuclear protein presence. (**B**) The yeast one-hybrid experiment indicates the binding of VuWRKY2 to the promoter of CAT. The promoter containing the TTGACT(C) element was cloned into the pLaczi vector, and VuWRKY2 was cloned into the pB42AD vector. Subsequently, the pLaczi vector and pB42AD-VuWRKY2 were co-transformed into the yeast strain EGY48. Streak the yeast transformants onto SD/-Ura-Trp plates with or without 20 mg/mL X-gal. pB42AD-p53 and pLaczi-p53 were used as positive controls. The red star represents the target protein.

## Data Availability

The transcriptomic data reported in this paper have been deposited in the Genome Sequence Archive (Genomics, Proteomics, and Bioinformatics 2021) in the National Genomics Data Center (Nucleic Acids Res 2022), China National Center for Bioinformation/Beijing Institute of Genomics, Chinese Academy of Sciences (GSA: CRA014681) that are publicly accessible at https://ngdc.cncb.ac.cn/gsa, accessed on 29 January 2024.

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
