# Peer review of "Identification and Analysis of WRKY Transcription Factors in Response to Cowpea Fusarium Wilt in Cowpea"

_plants, 2024, doi:10.3390/plants13162273_

Round 1
Reviewer 1 Report (Previous Reviewer 3)
Comments and Suggestions for Authors
After careful consideration and second review, I am pleased to accept the manuscript for publication.in the current form. The authors have addressed the majority of my concern in the first version.
Author Response
After careful consideration and second review, I am pleased to accept the manuscript for publication.in the current form. The authors have addressed the majority of my concern in the first version.
Response: Thanks for your suggestions. We greatly appreciate the reviewer’s meticulous work and patience in helping us improve the manuscript. We also thank you for recognizing our work.
Reviewer 2 Report (Previous Reviewer 1)
Comments and Suggestions for Authors
All my questions have been thoroughly answered, and I have no further inquiries at this time.
Comments on the Quality of English LanguageThe quality of the English language can be improved.
Author Response
All my questions have been thoroughly answered, and I have no further inquiries at this time.
Response: Thanks for your suggestions. We greatly appreciate the reviewer’s meticulous work and patience in helping us improve the manuscript. We also thank you for recognizing our work.
Reviewer 3 Report (New Reviewer)
Comments and Suggestions for Authors
The manuscript entitled " Identification and Analysis of WRKY Transcription Factors in Response to Cowpea Fusarium Wilt in Cowpea.", is focused on genome-wide identification of WRKY genes and their role in response to Cowpea Fusarium Wilt infection. The manuscript has the potential to make a significant contribution to the understanding of the function of some differentially expressed WRKY genes in Cowpea Fusarium Wilt. However, before making a final decision, I have suggestions that need to be slightly changes:
1) English language edit is must
2) Please explain the phrase:
Line 267, “a downward trend in expression”
Line 270, “swift activation of immune defense responses”
Line 360, “deformations like WRKYGKK”.
3) Line 279, the word “TPM” has but not with its full form.
Comments on the Quality of English LanguageEditing of English language is must.
Author Response
The manuscript entitled " Identification and Analysis of WRKY Transcription Factors in Response to Cowpea Fusarium Wilt in Cowpea.", is focused on genome-wide identification of WRKY genes and their role in response to Cowpea Fusarium Wilt infection. The manuscript has the potential to make a significant contribution to the understanding of the function of some differentially expressed WRKY genes in Cowpea Fusarium Wilt.
Response: Thanks for your suggestions. We appreciate the reviewer's recognition of our research content. We also greatly appreciate and agree with the suggestions for improving the manuscript. Below, please find our responses to all questions and comments.
However, before making a final decision, I have suggestions that need to be slightly changes:
1) English language edit is must
Response: Thank you for your suggestions. We have invited a native speaker to thoroughly revise the manuscript, ensuring smooth, accurate, and consistent writing. The revisions in the manuscript have been marked in green.
2) Please explain the phrase:
Line 267, “a downward trend in expression”
Response: Thank you for your suggestions. The phrase " a downward trend in expression" means that after 12 days of inoculation, the expression levels of the VuWRKY20 and VuWRKY52 genes in the treatment group were downregulated compared to the control group.
Line 270, “swift activation of immune defense responses”
Response: Thank you for your suggestions. In the manuscript, the phrase "swift activation of immune defense responses" means that the genes VuWRKY2, VuWRKY22, VuWRKY37, VuWRKY41, VuWRKY49, and VuWRKY70 showed significant upregulation in expression after being infected by the cowpea Fusarium wilt pathogen, compared to the control group. This change indicates that the plant's immune system rapidly recognized the threat and initiated defense mechanisms to resist pathogen invasion.
Line 360, “deformations like WRKYGKK”.
Response: Thank you for your suggestion. The expression "variants like WRKYGKK" is more appropriate than "deformations like WRKYGKK". This has been corrected in lines 355 to 356 of the manuscript. It refers to the WRKY transcription factor family, where the WRKY domain contains a conserved WRKYGQK motif at its N-terminus. Variants of the WRKYGQK heptapeptide sequence include WRKYGKK, WRKYGEK, WRKYGSK, or WRKYDQK.
3) Line 279, the word “TPM” has but not with its full form.
Response: Thank you for your suggestions. Based on your suggestion, we have added the full form of TPM as "Transcript per million" in the manuscript and highlighted it in red.
This manuscript is a resubmission of an earlier submission. The following is a list of the peer review reports and author responses from that submission.
Round 1
Reviewer 1 Report
Comments and Suggestions for Authors
The manuscript by Hao et al. analyzed 91 cowpea WRKY family genes through bioinformatic and limited experimental analysis. The novelty of this manuscript is very low, and the results provided are overly descriptive and lack clear conclusions. There are also notable deficiencies in the experimental design.
1. WRKY family genes are extensively proven to be involved in plant immunity. Key members such as WRKY33, WRKY75, WRKY48, WRKY50, and WRKY51 have been shown to be functionally important in Arabidopsis response to pathogens, as the authors introduced. Hao et al. performed the phylogenetic analysis of WRKY proteins from cowpea and Arabidopsis, but the information about the homologs of these functional genes in cowpea is absent.
2. In Figure 6A, the time points of 0 days and 5 days should also be included, and the scale bar should be added. Besides showing a representative figure, the physiological index should be included to reflect the degree of pathogen infection. In Figure 6B, the negative control without infection should be included, especially considering that the authors measured VuWRKY gene expression levels over many days. Transcriptional changes caused by developmental processes should be considered or excluded.
3. Regarding the few VuWRKY genes that the authors selected for RT-PCR analysis, except for indicating that they may respond to CFW infection, there is no data to indicate their function in response to CFW infection.
4. In Figure 7, there is only one fluorescence signal in all experimental groups, and some of them are challenging to identify as a nucleus. Colocalization with nuclear marker genes or DAPI staining should be considered.
Comments on the Quality of English Languagecan be improved.
Author Response
The manuscript by Hao et al. analyzed 91 cowpea WRKY family genes through bioinformatic and limited experimental analysis. The novelty of this manuscript is very low, and the results provided are overly descriptive and lack clear conclusions. There are also notable deficiencies in the experimental design.
Response: Thanks for your suggestions. We appreciate the reviewer's evaluation of our manuscript. Although the WRKY family has been extensively studied, the role of the cowpea WRKY family in pathogen infection, especially in response to Fusarium oxysporum infection, has not been reported to date. We have also revised and improved our experiments. We appreciate and agree with the suggestion to improve the manuscript. Below, please find our responses to all questions and comments.
1.WRKY family genes are extensively proven to be involved in plant immunity. Key members such as WRKY33, WRKY75, WRKY48, WRKY50, and WRKY51 have been shown to be functionally important in Arabidopsis response to pathogens, as the authors introduced. Hao et al. performed the phylogenetic analysis of WRKY proteins from cowpea and Arabidopsis, but the information about the homologs of these functional genes in cowpea is absent.
Response: Thanks for your suggestions. In response to your comment on the lack of information on functional gene homologs in cowpea, we first supplemented the information with representatives from the classification of the WRKY family in Arabidopsis thaliana. Additionally, we identified the entire WRKY gene family in Glycine max, another legume, and analyzed the evolutionary relationships between the WRKYs in Arabidopsis thaliana, Glycine max, and VuWRKY. The phylogenetic tree results indicate that VuWRKYs are closely related to the WRKYs in Arabidopsis thaliana and Glycine max. The homologous genes of WRKY33 are VuWRKY84 and VuWRKY38. In Arabidopsis, AtWRKY33 can regulate the expression of many disease resistance-related genes. Furthermore, AtWRKY33 is induced by pathogen infection, thereby enhancing resistance to Alternaria brassicicola and Botrytis cinerea. The homologous gene of WRKY75 is VuWRKY2. In Arabidopsis, AtWRKY75 can regulate the JA-mediated signaling pathway, positively regulating plant resistance to Botrytis cinerea. The homologous gene of WRKY48 is VuWRKY51. In Arabidopsis, AtWRKY48 has been proven to be a negative regulator of basal defense against the bacterial pathogen Pseudomonas syringae, functioning by negatively regulating the expression of pathogen-induced pathogenesis-related genes. The homologous gene of WRKY50 is VuWRKY81, and the homologous genes of WRKY51 are VuWRKY16 and VuWRKY75. In Arabidopsis, both AtWRKY50 and AtWRKY51 can positively regulate resistance to Phytophthora infestans through the SA-mediated signaling pathway. Among them, the homologous genes VuWRKY2, VuWRKY16, VuWRKY75, and VuWRKY81 are differentially expressed under Fusarium wilt stress, indicating their response to Fusarium wilt infection. It is speculated that these VuWRKY genes may also have functions similar to their homologous genes in Arabidopsis. We have included the aforementioned results as Figure S1; additionally, we have added the results of the homology analysis to our discussion (Line 387-404).
2.In Figure 6A, the time points of 0 days and 5 days should also be included, and the scale bar should be added. Besides showing a representative figure, the physiological index should be included to reflect the degree of pathogen infection. In Figure 6B, the negative control without infection should be included, especially considering that the authors measured VuWRKY gene expression levels over many days. Transcriptional changes caused by developmental processes should be considered or excluded.
Response: Thanks for your suggestions. According to your suggestion, we have added photos taken at 0-day and 5-day time points to Figure 6A, including scale bars and corresponding day labels. Additionally, we have conducted disease index statistics for cowpea plants post-inoculation, which have been included in Table S4.
Regarding your suggestion that Figure 6B lacked a negative control, we have added a negative control to the figure to exclude transcriptional changes caused by developmental processes.
3.Regarding the few VuWRKY genes that the authors selected for RT-PCR analysis, except for indicating that they may respond to CFW infection, there is no data to indicate their function in response to CFW infection.
Response: Thanks for your suggestions. The current purpose of this experiment is to find the WRKY gene that responds to cowpea wilt pathogen infection. These preliminary results will lay the foundation for further functional validation. We plan to conduct more in-depth studies on their roles in responding to cowpea wilt pathogen infection, such as gene silencing and overexpression.
4.In Figure 7, there is only one fluorescence signal in all experimental groups, and some of them are challenging to identify as a nucleus. Colocalization with nuclear marker genes or DAPI staining should be considered.
Response: Thanks for your suggestions. According to your suggestion, we have supplemented the DAPI staining experiment in Figure 7. The fluorescence signals overlapped with DAPI nuclear staining, providing more convincing evidence that the four VuWRKY proteins are nuclear-localized.

Reviewer 2 Report
Comments and Suggestions for Authors
The authors in their manuscript entitled “Identification and Analysis of WRKY Transcription Factors in Response to Cowpea Fusarium Wilt in Cowpea” present a molecular approach for the characterization of WRKY TFs in the frame of the Cowpea- Fusarium (plant-pathogen) interactions.
The aim is clear, the methodological approach is standard, and its implementation is correct. The presentation of the results is clear and straight-forward and their discussion does not require further argumentation. There are no major comments on the presented work.
The only concept that requires further discussion is the reasoning for the selection of the specific genes to study in qPCR (line 245) and the choice of WRKY 2, 22, 49 and 59 genes for the localization study. The authors could provide a related reasoning for both since other genes could have been chosen for gene expression (e.g. 27, 42 or 76) and many other genes have a predicted nuclear localization as well. A related argumentation could also give reasoning for commenting on these specific/selected genes in the Discussion part as well.
Please also comment on the second (visible) band in the WRKY22-GFP lane in figure 8A. Is there an idea why this appears in the W. blot?
Author Response
The authors in their manuscript entitled “Identification and Analysis of WRKY Transcription Factors in Response to Cowpea Fusarium Wilt in Cowpea” present a molecular approach for the characterization of WRKY TFs in the frame of the Cowpea- Fusarium (plant-pathogen) interactions.
The aim is clear, the methodological approach is standard, and its implementation is correct. The presentation of the results is clear and straight-forward and their discussion does not require further argumentation. There are no major comments on the presented work.
Response: Thanks for your suggestions. We appreciate the reviewer's recognition of our work's clear research objectives and clear presentation of results. We also appreciate and agree with the suggestion to improve the manuscript. Below, please find our responses to all questions and comments.
The only concept that requires further discussion is the reasoning for the selection of the specific genes to study in qPCR (line 245) and the choice of WRKY 2, 22, 49 and 59 genes for the localization study. The authors could provide a related reasoning for both since other genes could have been chosen for gene expression (e.g. 27, 42 or 76) and many other genes have a predicted nuclear localization as well. A related argumentation could also give reasoning for commenting on these specific/selected genes in the Discussion part as well.
Response: Thanks for your comments. We detected a total of 31 differentially expression cowpea WRKY genes across three different groups by transcriptional analysis of 91 cowpea WRKY family members following inoculation with cowpea Fusarium wilt pathogen. The criteria for selecting the target genes for the next experiment are these genes come from three different groups, and their orthologous genes are involved in the response to biotic stress. In total, we carefully selected 12 VuWRKY genes. However, we failed to detect the expression of VuWRKY27, resulting in the final selection of 11 genes. The main reason for selecting these 4 WRKY genes (WRKY2, 22, 49, and 59) for nuclear localization experiments is that the biotic-stress-ressitant function of their orthologous genes has been reported [1-5], and we want to further study the functions of these 4 genes.
References
- CHEN L G, ZHANG L P, XIANG S Y, CHEN Y L, ZHANG H Y, YU D Q. The transcription factor WRKY75 positively regulates jasmonate-mediated plant defense to necrotrophic fungal pathogens. Journal of Experimental Botany. 2021, 72, 1473-1489
- Encinas-Villarejo S, Maldonado AM, Amil-Ruiz F, de los Santos B, Romero F, Pliego-Alfaro F, Muñoz-Blanco J, Caballero JL. Evidence for a positive regulatory role of strawberry (Fragaria x ananassa) Fa WRKY1 and Arabidopsis AtWRKY75 proteins in resistance. J Exp Bot. 2009, 60, 3043-65.
- Lai ZB, Vinod KM, Zheng ZY, Fan B.F, and Chen ZX. Roles of Arabidopsis WRKY3 and WRKY4 transcription factors in plant responses to pathogens, BMC Plant Biol. 2008, 8, 68.
- Zheng ZY, Mosher SL, Fan BF, Klessig DF, and Chen ZX. Functional analysis of Arabidopsis WRKY25 transcription factor in plant defense against Pseudomonas syringae. BMC Plant Biol. 2007, 7, 2.
- Xu X, Chen C, Fan B, and Chen Z. Physical and func tional interactions between pathogen-induced Arabidopsis WRKY18, WRKY40, and WRKY60 transcription factors. Plant Cell, 2006, 18, 1310-1326.
Please also comment on the second (visible) band in the WRKY22-GFP lane in figure 8A. Is there an idea why this appears in the W. blot?
Response: Thanks for your suggestions. In Figure 8A, additional bands can be observed in the channel for VuWRKY22-GFP, apart from the target band. We conducted this experiment three times, and each time detected these bands, ruling out protein degradation as the cause. Post-transcriptional and translational modifications are the most common factors causing such bands. After transcription, selective splicing can lead to the translation of proteins of different sizes from the same transcript, known as splice variants of the protein. Therefore, we speculate that the band is derived from protein degradation, post transcriptional regulation or splice variants of the protein, which more work is needed.

Reviewer 3 Report
Comments and Suggestions for Authors
The m/s by Hao et al identified 91 WRKY genes in cowpea, analyzing their expression patterns in response to Fusarium wilt. The authors analysed the Phylogenetic analysis classified these genes into three groups, with specific variations in conserved protein domains. Gene expression analysis revealed differential expression of VuWRKY genes under Fusarium wilt stress, with four genes localized in the cell nucleus. The paper is well written and demonstrates a commendable level of thoroughness and significance on further identifying and analysing WRKY Tfs in cowpea response to Fusarium wilt. Your findings shed light on the complex regulatory mechanisms underlying cowpea's defense against this devastating pathogen. Pending major revisions to address some additional data and clarification points, I am inclined to accept this paper. Overall, the study represents a valuable contribution to our understanding of plant-pathogen interactions and holds considerable promise for future research in crop protection and breeding/GE strategies
Authors hsould address these weaknesses before proceeding with the next stage:
- introduction provides a good background on WRKY genes and their importance in plant responses to stress, however it lacks clarity of theis research objectives and hypotheses (specific gaps).
- Results
- Authors should validate the interaction VuWRKY2-CAT promoter through additional assays, (ChIP orY2H assays).
- Did the authors try to identify (or identified) the functional implications of variations in conserved protein domains in certain VuWRKY genes ?. Additional data on the in vitro DNA binding assays would be helpful to identify these conserved proteins.
- The weakest part of this paper is the Discussion section; it lacks depth in interpreting the results in the context of existing literature. It could be strengthened by providing more extensive comparisons with previous studies on WRKY genes in other plant species and discussing the implications of the current findings for understanding cowpea Fusarium wilt resistance.
Additionally, while the study identifies potential mechanisms underlying cowpea's response to Fusarium wilt, it does not propose specific hypotheses or directions for future research. Including suggestions for further investigations would enhance the significance of the study's conclusions.
By addressing incorporating the suggested improvements, the paper can enhance its clarity, significance, and contribution to the understanding of WRKY gene regulation in cowpea Fusarium wilt resistance.
Comments on the Quality of English LanguageMinor editing of English language required
Author Response
The m/s by Hao et al identified 91 WRKY genes in cowpea, analyzing their expression patterns in response to Fusarium wilt. The authors analysed the Phylogenetic analysis classified these genes into three groups, with specific variations in conserved protein domains. Gene expression analysis revealed differential expression of VuWRKY genes under Fusarium wilt stress, with four genes localized in the cell nucleus. The paper is well written and demonstrates a commendable level of thoroughness and significance on further identifying and analysing WRKY Tfs in cowpea response to Fusarium wilt. Your findings shed light on the complex regulatory mechanisms underlying cowpea's defense against this devastating pathogen. Pending major revisions to address some additional data and clarification points, I am inclined to accept this paper. Overall, the study represents a valuable contribution to our understanding of plant-pathogen interactions and holds considerable promise for future research in crop protection and breeding/GE strategies
Response: Thanks for your suggestions. We appreciate the reviewer's recognition of our work. We also appreciate and agree with the suggestion to improve the manuscript. Below, please find our responses to all questions and comments.
Authors hsould address these weaknesses before proceeding with the next stage:
- introduction provides a good background on WRKY genes and their importance in plant responses to stress, however it lacks clarity of theis research objectives and hypotheses (specific gaps).
Response: Thanks for your suggestions. Based on your suggestion that the introduction lacks hypotheses, we have made modifications and added this content in the introduction (Line105-118): Although the identification of WRKY transcription factors in cowpea has been reported, there have been no reports so far on the role of the cowpea WRKY family in response to pathogen infection, particularly Fusarium oxysporum infection. This study identified and analyzed the expression of potential WRKY genes using the whole genome of cowpea as a reference (the cowpea genome size is 518.6 Mb with a total of 11 chromosomes). There are 31 differentially expressed genes in response to the cowpea Fusarium wilt, among which 4 genes, as confirmed by subcellular localization and Western blot experiments, were found to be localized in the cell nucleus and expressed normally in N. benthamiana leaves. VuWRKY2 binding to the promoter region of the CAT gene suggests its potential involvement in transcriptional regulation. These findings lay the groundwork for further research into the mechanisms underlying cowpea's response to Fusarium wilt. This provides data support and theoretical basis for studying the response mechanism of this transcription factor in cowpea under Fusarium wilt infection and stress conditions, as well as its role in biotic stress defense response and functional verification.
- Results
- Authors should validate the interaction VuWRKY2-CAT promoter through additional assays, (ChIP orY2H assays).
Response: Thanks for your suggestions. Your suggestions regarding the use of ChIP and Y2H experiments to validate the interaction between VuWRKY2 and the CAT promoter are very valuable. But the current purpose of this experiment is to find the WRKY gene that responds to cowpea wilt pathogen infection. These preliminary results will lay the foundation for further functional validation in future studies, according to your suggestion, we will implement these validation methods to better understand the response mechanisms of cowpea to pathogen infection.
- Did the authors try to identify (or identified) the functional implications of variations in conserved protein domains in certain VuWRKY genes? Additional data on the in vitro DNA binding assays would be helpful to identify these conserved proteins.
Response: Thanks for your suggestions. The idea you mentioned about performing point mutations on certain WRKY domain sequences and then observing if these mutations affect the binding of the WRKY domain to the target DNA is very meaningful. However, the primary goal of our current experiment is to identify WRKY genes that respond to cowpea wilt pathogens and potential target genes. In the next step, when delving into the VuWRKY2-CAT interaction, we will consider using point mutations to study the functional interaction of VuWRKY2-CAT as per your suggestion.
- The weakest part of this paper is the Discussion section; it lacks depth in interpreting the results in the context of existing literature. It could be strengthened by providing more extensive comparisons with previous studies on WRKY genes in other plant species and discussing the implications of the current findings for understanding cowpea Fusarium wilt resistance.
Additionally, while the study identifies potential mechanisms underlying cowpea's response to Fusarium wilt, it does not propose specific hypotheses or directions for future research. Including suggestions for further investigations would enhance the significance of the study's conclusions.
By addressing incorporating the suggested improvements, the paper can enhance its clarity, significance, and contribution to the understanding of WRKY gene regulation in cowpea Fusarium wilt resistance.
Response: Thanks for your suggestions. We have diligently supplemented and refined the content of our discussion (Line387-417), with specific additions such as: In this study, it was found that the expression levels of 31 VuWRKY genes changed after infection by the cowpea Fusarium wilt pathogen, indicating their involvement in the response to Fusarium wilt stress in cowpea. However, the expression intensity of these 31 differentially expressed VuWRKY genes varied, and many of these genes have homologous genes involved in plant immune responses. For example, the homologous gene of VuWRKY2, AtWRKY75, may form complexes with ET signaling or other defense-related proteins to regulate defense responses against necrotrophic pathogens [49]. AtWRKY75 positively regulates Sclerotinia sclerotiorum [50]. The homologous genes of VuWRKY22, AtWRKY4, and AtWRKY3 play positive regulatory roles in the resistance response to saprophytic pathogens, with single mutants and double mutants showing reduced resistance to Botrytis cinerea. However, unlike others, AtWRKY4 overexpression plants are also more susceptible to Pseudomonas syringae [51]. The homologous gene of VuWRKY49, AtWRKY25, can negatively regulate resistance to Pseudomonas syringae [52]. The double mutants of VuWRKY59's homologous gene AtWRKY18 and AtWRKY60 show increased resistance to leaf spot pathogens and decreased resistance to Botrytis cinerea [53]. In this study, the cowpea Fusarium wilt pathogen, along with Sclerotinia sclerotiorum and Botrytis cinerea, are all fungi, indicating that the VuWRKY genes homologous to AtWRKY genes may have similar functions in response to fungal infections.
In this study, subcellular localization and Western blot analysis of VuWRKY2, VuWRKY22, VuWRKY49, and VuWRKY59 indicated that these four VuWRKY genes are located in the nucleus and are normally expressed in N. benthamiana leaves, with significant differential expression observed at the early stages of infection. The results suggest that the differential expression of VuWRKY genes is related to the response to cowpea Fusarium wilt infection. This is the first report of WRKY genes responding to cowpea Fusarium wilt infection in cowpea. Additionally, VuWRKY2 binds to the CAT promoter region, possibly participating in transcriptional regulation. These findings provide clues for further studies on the role of VuWRKY in the resistance mechanism of cowpea to Fusarium wilt. Therefore, this study enhances our understanding of how WRKY genes regulate plant responses to Fusarium wilt infection. However, further research is needed to elucidate the underlying mechanisms, laying the foundation for breeding cowpea for disease resistance.
References
49.CHEN L G, ZHANG L P, XIANG S Y, CHEN Y L, ZHANG H Y, YU D Q. The transcription factorWRKY75 positively regulates jasmonate-mediated plant defense to necrotrophic fungal pathogens. Journal of Experimental Botany. 2021, 72, 1473-1489
50.Encinas-Villarejo S, Maldonado AM, Amil-Ruiz F, de los Santos B, Romero F, Pliego-Alfaro F, Muñoz-Blanco J, Caballero JL. Evidence for a positive regulatory role of strawberry (Fragaria x ananassa) Fa WRKY1 and Arabidopsis At WRKY75 proteins in resistance. J Exp Bot. 2009, 60, 3043-65.
51.Lai ZB, Vinod KM, Zheng ZY, Fan B.F, and Chen ZX. Roles of Arabidopsis WRKY3 and WRKY4 transcription factors in plant responses to pathogens, BMC Plant Biol. 2008, 8, 68.
52.Zheng ZY, Mosher SL, Fan BF, Klessig DF, and Chen ZX. Functional analysis of Arabidopsis WRKY25 transcription factor in plant defense against Pseudomonas syringae. BMC Plant Biol. 2007, 7, 2.
53.Xu X, Chen C, Fan B, and Chen Z. Physical and func tional interactions between pathogen-induced Arabidopsis WRKY18, WRKY40, and WRKY60 transcription factors. Plant Cell, 2006, 18, 1310-1326.
